# DSA: Efficient Serving For Video Generation Models via Distributed Sparse Attention

**Shenggui Li**
Nanyang Technological University
shenggui001@e.ntu.edu.sg

**Runyu Lu**
University of Michigan
runyulu@umich.edu

**Qiaolin Chen**
Nanyang Technological University
qiaoling.chen@ntu.edu.sg

**Haiyan Yin**
CFAR and IHPC, Agency for Science,
Technology and Research (A*STAR), Singapore
yin_haiyan@a-star.edu.sg

**Yueming Lyu**
CFAR and IHPC, Agency for Science,
Technology and Research (A*STAR), Singapore
lyu_yueming@a-star.edu.sg

**Yonggang Wen**
Nanyang Technological University
ygwen@ntu.edu.sg

**Ivor Tsang**
CFAR and IHPC, Agency for Science,
Technology and Research (A*STAR), Singapore
ivor_tsang@a-star.edu.sg

**Tianwei Zhang**
Nanyang Technological University
tianweizhang@ntu.edu.sg

## ABSTRACT

Diffusion Transformer models have driven the rapid advances in video generation, achieving state-of-the-art quality and flexibility. However, their attention mechanism remains a major performance bottleneck, as its dense computation scales quadratically with the sequence length. To overcome this limitation and reduce the generation latency, we propose DSA, a novel attention mechanism that integrates sparse attention with distributed inference for diffusion-based video generation. By leveraging carefully-designed parallelism strategies and scheduling, DSA significantly reduces redundant computation while preserving global context. Extensive experiments on benchmark datasets demonstrate that, when deployed on 8 GPUs, DSA achieves up to 1.43× inference speedup than the existing distributed method and 10.79× faster than single-GPU inference.

## 1 INTRODUCTION

Recent advances in generative models have transformed the landscape of digital content creation, introducing unprecedented capabilities in generating sophisticated visual content (Rombach et al., 2021; Ho et al., 2020; Song et al., 2020; Ding et al., 2021). This breakthrough has streamlined creative processes across multiple industries, from artistic design to media production. Particularly, advanced video generative models have been integrated into professional workflows through proprietary commercial platforms such as Google Veo, Kwai Kling and OpenAI Sora, as well as open-sourced alternatives like Stable Video Diffusion (Blattmann et al., 2023), Mochi (Team, 2024), CogVideo (Hong et al., 2023), Hunyuan Video (Kong et al., 2024) and Wan (Wan et al., 2025).

In the field of vision generation, diffusion transformer models (DiTs) have emerged as a cornerstone, renowned for their ability to synthesize highly realistic and visually coherent outputs (Peebles & Xie, 2022). By setting new benchmarks in video quality, these models represent a major step forward in computer-generated content. However, this advantage comes at the cost of prohibitive inference latency due to the substantial computational overhead of the attention mechanism. In practice, DiTs often rely on full attention across temporal and spatial dimensions (Zheng et al., 2024; Lin et al.,

2024; Hong et al., 2023; Wan et al., 2025), which incurs quadratic complexity with respect to the sequence length. This scaling bottleneck poses severe challenges for generating high-resolution, long-duration videos. For instance, producing a 5-second, 720p video with Wan2.1-14B (Wan et al., 2025) requires approximately 31 minutes. This underscores the inefficiency of current approaches and their prohibitive nature for commercial applications, necessitating further optimization.

Prior projects focus on the transformation from dense attention to sparse attention (Zhang et al., 2025a; Xi et al., 2025; Zhang et al., 2025c;b). Video data inherently exhibit sparsity in the temporal and spatial dimension. Therefore, sparse attention methods typically rely on the observation that only a subset of temporal or spatial tokens contribute significantly to the next-step denoising. By dynamically pruning attention maps, these methods achieve notable FLOP reductions without retraining. However, such savings alone are insufficient at scale. Another domain focuses on the system optimization. xDiT (Fang et al., 2024) successfully applies sequence parallelism (Li et al., 2023; Fang & Zhao, 2024; Liu et al., 2024; Jacobs et al., 2024) for video generations. By splitting the hidden states along the sequence dimension, sequence parallelism can evenly distribute the computation workloads across GPUs, reducing the overall latency. However, this method often achieves sub-linear scaling due to extra communication overhead. One direction for further improvement is the integration of sparse attention and distributed inference. MagiAttention (Zewei & Yunpeng, 2025) combines sparse attention and distributed attention. However, it is used for training Large Language Models (LLMs) instead of inference.

Our proposed Distributed Sparse Attention (DSA) bridges this gap by jointly exploiting redundancy in attention maps and the computational capacity of distributed hardware. DSA is built on two key components: mixed parallelism (MP) and dynamic attention scheduling (DAS). At runtime, a lightweight profiler determines the attention pattern for each head, after which the most suitable sequence parallelism strategy is applied. This adaptive choice ensures that both computation and communication overheads are substantially reduced. Furthermore, since the distribution of attention patterns can vary across layers and time steps, DAS dynamically adjusts the execution order to better overlap computation with communication, thereby maximizing the efficiency. Notably, this design achieves super-linear scaling, enabling larger models to run faster than their smaller counterparts under specific configurations.

Overall, our work makes the following key contributions: (1) We analyze the runtime characteristics of advanced DiT models during video generation and identify the computation bottleneck. (2) We propose DSA, a novel training-free attention mechanism which integrates both sparse attention and distributed inference. (3) We conduct extensive experiments to evaluate DSA, demonstrating its ability to reduce end-to-end latency by $11\times$ while maintaining the video quality.

## 2 PRELIMINARIES

### 2.1 DIFFUSION

Diffusion models are based on a stochastic denoising process, where data is gradually corrupted by noise via a forward diffusion process and then reconstructed using a learned reverse process (Rombach et al., 2021; Song et al., 2020; Ho et al., 2020). The forward process is defined as:

$$q(x_t|x_{t-1}) = \mathcal{N}(x_t; \sqrt{\alpha_t}x_{t-1}, (1 - \alpha_t)I)$$

where $x_t$ represents the noisy data at timestep $t$, and $\alpha_t$ controls the variance schedule. The reverse process is parameterized by a neural network $\epsilon_\theta$, which predicts the noise added at each timestep. The reverse transitions are modeled as:

$$p_\theta(x_{t-1}|x_t) = \mathcal{N}(x_{t-1}; \mu_\theta(x_t, t), \Sigma_\theta(x_t, t))$$

where $\mu_\theta$ and $\Sigma_\theta$ are learned mean and variance functions. The model iteratively refines a noisy sample until it converges to the original data distribution.

Diffusion models excel in their ability to handle complex data distributions and produce high-resolution outputs, making them a preferred choice for generative tasks. However, their iterative denoising process requires multiple forward passes through the network, resulting in high computational and memory demands.

## 2.2 DIFFUSION TRANSFORMER

Transformers, originally designed for sequence-to-sequence tasks in natural language processing (Vaswani et al., 2017), become a cornerstone of modern AI architectures. Their self-attention mechanism enables effective modeling of long-range dependencies, making them well-suited for diverse tasks, including generative modeling (Liu et al., 2021; Dosovitskiy et al., 2020; Peebles & Xie, 2022). In recent diffusion models, transformers are often employed as the backbone for the denoising network, where they learn to predict the noise or original data distribution at each timestep.

The transformer architecture relies on self-attention (MHSA) layers and feedforward neural networks. The self-attention mechanism computes a weighted representation of input tokens by attending to their pairwise relationships:

$$\texttt{Attention}(Q, K, V) = \texttt{softmax}(\frac{QK^T}{\sqrt{d_k}})V$$

where $Q$, $K$, and $V$ represent the query, key, and value matrices, and $d_k$ is the dimensionality of the keys. The attention module can capture global context efficiently, which is critical for vision tasks.

## 2.3 SPARSE ATTENTION

Sparse attention exploits the fact that only a subset of tokens—either within frames or across time—contribute significantly to the output, allowing many attention computations to be skipped. Broadly, sparse attention can be categorized into static and dynamic patterns. Static sparse attention relies on predefined attention masks, typically designed based on observed runtime characteristics of the model. Because the computation pattern is fixed in advance, it enables the use of high-performance kernels. In contrast, dynamic sparse attention determines the sparse patterns on the fly during inference, usually by approximating query–key interactions. While static patterns offer efficiency through predictable computation, dynamic patterns provide greater adaptivity.

Examples of static sparse attention include MInference (Jiang et al., 2024), STA (Zhang et al., 2025c), and SVG (Xi et al., 2025). Among them, SVG achieves the best performance, as it preserves the original video generation quality without degradation. In contrast, dynamic sparse attention is exemplified by SpargeAttention (Zhang et al., 2025a), which pools query and key tokens and computes cosine similarities to identify critical attention blocks in an online manner, skipping the unimportant ones. SpargeAttention is versatile and can be applied to large language models, image generation models, and video generation models. However, its performance lags behind state-of-the-art static methods such as SVG, particularly in maintaining video quality.

## 2.4 SEQUENCE PARALLELISM

Traditional parallelism strategies, including data, tensor and pipeline parallelism (Li et al., 2020; Zheng et al., 2022; Rasley et al., 2020; Narayanan et al., 2021), do not scale well when the sequence length becomes extremely large. Sequence parallelism (SP) partitions the input along the sequence dimension across devices to distribute both memmory and compute burden for attention over long sequences. There are mainly three categories of sequence parallelism:

- SP-Ring (Li et al., 2023; Liu et al., 2024): The sequence data is partitioned into chunks and distributed across devices in a ring layout. During the attention operation, the key and value embeddings are circulated among devices in a ring fashion via peer-to-peer (P2P) communication, which is often overlapped with computation to improve efficiency.
- SP-Ulysses (Jacobs et al., 2023): The input is also partitioned along the sequence dimension. Through an all-to-all communication step, these chunks are redistributed so that each GPU holds the full sequence for a subset of attention heads. Local attention is computed independently for each head, after which the outputs are redistributed to restore the original sequence partitioning.
- SP-Unified (Fang & Zhao, 2024; Gu et al., 2024):This is a hybrid sequence parallelism, combining the strengths of Ulysses and Ring while mitigating their respective limitations. Devices are organized into a two-dimensional grid (mesh): Ulysses is applied along one dimension (rows), while Ring is applied along the other (columns). Redistribution via all-to-all and send-receive communication ensures proper transfer of data between sequence partitions and head slices.

## 3 CHALLENGES AND MOTIVATION

Deploying a transformer model for video generation has the following challenges.

**1. Attention is the computational bottleneck.** A high-resolution video is typically flattened into a long sequence of vision tokens. Taking Wan-2.1-14B as an example, a 5-second 720p video corresponds to approximately 302k tokens per input channel, with a total of 16 channels. As the attention module scales quadratically with the sequence length, it accounts for a substantial fraction of inference time, evidenced by the breakdown of inference execution time for Wan2.1-14B model with flash attention (Dao, 2024) in Figure 1. This overhead becomes even more pronounced when scaling to longer durations or higher resolutions.


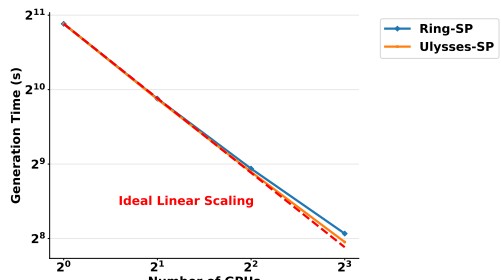

Figure 1: Execution time breakdown of 720p 5-second video generation of different models on H100 GPUs

Figure 2: Weak scaling of a 720p 5-second video using Wan2.1-14B on H100 GPUs, showing sub-linear decrease in generation time

**2. Sparse attention is not scalable.** Since high-resolution videos lead to long sequences of vision tokens for inference, it is natural to adopt parallel inference strategies such as sequence parallelism to reduce per-device computational overhead. However, existing sparse attention methods are not designed for multi-GPU inference and thus fail to scale efficiently. Sequence parallelism partitions the token sequence into sub-chunks, with each device responsible for a subset of the query, key, and value embeddings. A key challenge arises because existing sparse attention methods (Xi et al., 2025; Zhang et al., 2025a;c) require access to the full-sequence query and key to determine the sparse attention pattern. Under ring-style sequence parallelism (Li et al., 2023; Liu et al., 2024), this leads to excessive communication overhead as devices must exchange full embeddings. Ulysses-style sequence parallelism (Jacobs et al., 2024) alleviates this by gathering embeddings via all-to-all communication, but still incurs significant overhead since tokens outside the sparse mask are redundantly transferred.

Meanwhile, existing sparse attention methods fail to consider additional complexities when scaling sparse attention to distributed settings. For instance, attention sinks (Xiao et al., 2024b) can be observed in video generation models (Xi et al., 2025). When distributing the sequence over GPUs, only 1 GPU holds attention sink tokens, while these tokens need to be attended by all other GPUs.

**3. Sequence parallelism is sub-linearly scalable.** Sequence parallelism is an effective approach for handling long-sequence training and inference. However, it introduces additional communication overhead since query, key, and value embeddings must be exchanged across devices. Consequently, the scaling efficiency becomes sub-linear, meaning adding more GPUs does not yield a proportional reduction in latency. As shown in Figure 2, when generating a 5-second 720p video using Wan2.1-14B on H100 GPUs, the inference time only reduces from 1837.9s to 287.9s when scaling from 1 to 8 GPUs, equivalent to a scaling efficiency of 79.7%. Thus, sequence parallelism only trades the overall throughput for a single request latency. This limitation raises concerns about the cost-effectiveness of sequence parallelism in commercial model-as-a-service (MaaS) deployments.

## 4 DISTRIBUTED SPARSE ATTENTION

To address the above challenges, we introduce Distributed Sparse Attention (DSA), a methodology that integrates sparse attention with distributed inference for efficient video generation. In contrast to computing full-attention with sequence parallelism, DSA selectively matches the sparse pattern and

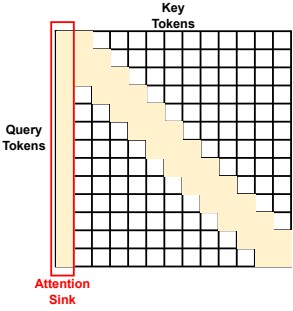 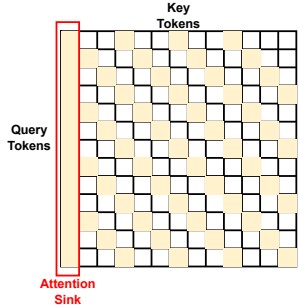

(a) Spatial attention pattern          (b) Temporal attention pattern

Figure 3: Attention patterns which are unique to video generation models (Xi et al., 2025). In the spatial attention pattern, query tokens primarily attend to key tokens within the same frame or in adjacent frames, reflecting spatial locality. In contrast, the temporal attention pattern involves query tokens attending to key tokens at the same spatial location but across different frames. Both patterns exhibit the attention sink pattern (Xiao et al., 2024b), which all query tokens attend to the first few key tokens, which are often the text tokens in video generation.

parallel strategy, leading to significant lower computational overhead. During the communication of query, key and value embeddings, `DSA` can also filter out the unimportant tokens but only transfer critical tokens to the target device, reducing the overall communication overhead. As a result, `DSA` achieves sparse computation and super-linear scalability while preserving the video generation quality, leading to significant reduction in the generation latency and deployment cost.

## 4.1 SPARSITY PATTERN MATCHING

Existing methods, such as SVG (Xi et al., 2025), adopt static sparsity patterns for video generation models to achieve training-free inference acceleration. These static patterns are effective because video generation models exhibit distinct attention patterns, specifically spatial sparsity and temporal sparsity. Similar patterns have been observed in Large Language Models (Xiao et al., 2025). SVG pre-defines spatial and temporal sparse masks and matches attention heads to one of these masks through sampling, achieving video generation without quality loss.

While we adopt the same static pattern strategy, a challenge arises in distributed inference: video data is split into sub-sequences, rendering the previous matching strategy—which relies on full sequences—ineffective. To address this limitation, we employ local pattern matching with majority voting. We create pre-defined attention masks for local query and key sub-sequences, ensuring mask locations align with the corresponding query and key positions. Subsequently, we perform all-gather operations to aggregate local pattern matching decisions and vote on the final sparsity pattern.

## 4.2 MIXED PARALLELISM

Existing models, including Hunyuan-Video (Kong et al., 2024), Wan (Wan et al., 2025), and Step-Video (Ma et al., 2025), have adopted unified sequence parallelism (USP) (Fang & Zhao, 2024) as their default parallelization strategy. USP combines ring-style (Li et al., 2023; Liu et al., 2024) and Ulysses-style (Jacobs et al., 2023) sequence parallelism approaches. Specifically, it first performs all-to-all operations to gather sub-sequences, then executes ring-style attention to exchange key-value embeddings for self-attention computation. This design allows USP to degenerate to ring-style sequence parallelism when the Ulysses degree is 1, and vice versa.

However, this hybrid design is primarily optimized for cross-node communication. In contrast, model deployment is typically confined to a single node, since video generation models generally range from several billion to around 20 billion parameters. Furthermore, USP fails to account for the attention patterns inherent in video generation models. As demonstrated in prior work, attention maps in video generation models exhibit sparsity, particularly in the form of temporal and spatial sparse attention patterns illustrated in Figure 3. Current approaches lack specialized designs that leverage these distinct attention patterns to reduce the computational and communication overhead.

To address this limitation, we propose Mixed Parallelism (MP). As shown in Figure 3, the spatial and temporal patterns show distinctive features: the spatial sparsity occurs as the tokens are attending to its spatially close tokens in the same frame or in the nearby frames while the temporal sparsity shows that the tokens are attending other tokens at the same spatial location but across different frames. Thus, it can be wiser to apply a distinct parallel strategy to each sparsity pattern.

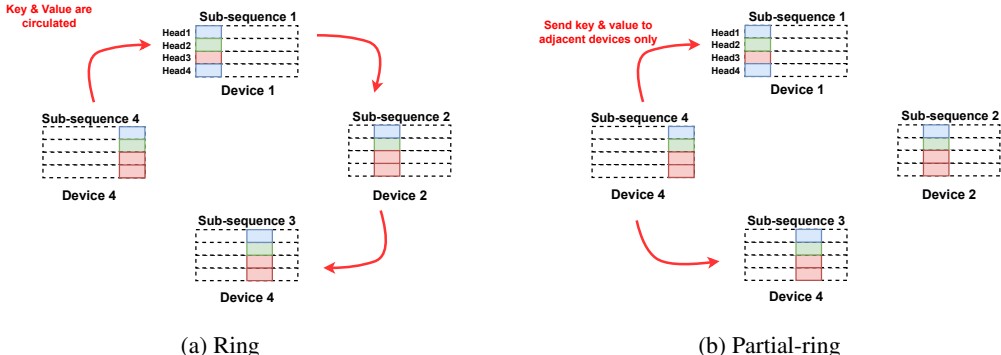

(a) Ring           (b) Partial-ring

Figure 4: Comparison between the original ring-style attention (a) and partial-ring attention (b). The typical ring attention will transfer the key and value embeddings from one device to others, resulting in $N - 1$ times of data transfer. By leveraging the spatial attention pattern, partial-ring only transfers the embeddings to the adjacent neighbors, keeping the number of data transfer to 2.

**Spatial Sequence Parallel.** This parallel strategy is applied to spatial sparse patterns. Given $N$ devices, each video sequence is partitioned into $N$ chunks of sub-sequences. Since query tokens primarily attend to spatially proximate key tokens in spatial sparsity, we can simplify sparse attention to local and adjacent computation only. However, this approach introduces two key complexities:

- Attention sink tokens: The first frame contains attention sink tokens that require global attention. Specifically, tokens in the first frame located on the first device must be attended by all other query tokens across devices.
- Variable spatial proximity ranges: The range of spatially proximate tokens varies across different attention heads. In some cases, spatial tokens located on adjacent devices also require attention.

To address these challenges, we broadcast attention sink tokens from the first device to all others and perform partial-ring communication for adjacent spatial tokens, as illustrated in Figure 4b. We compute attention outputs in chunks using online softmax (Dao, 2024) and overlap communication with computation. Since we only attend to spatially adjacent tokens, our approach performs send-receive operations only twice (one clockwise and one counterclockwise), compared to the $N - 1$ iterations required by typical ring attention. This design significantly reduces communication costs as the number of GPUs increases. Moreover, by incorporating adjacent spatial tokens rather than relying solely on local computation, we better preserve video generation quality.

**Temporal Sequence Parallel.** For temporal sparsity, the challenge is more complex due to repeated diagonal attention patterns that require query tokens on one device to attend to key tokens distributed across all devices. This necessitates the use of sequence parallelism. To achieve higher computational efficiency on modern accelerators such as GPUs, we perform layout transformations on temporal sparsity patterns to enable blockwise computation.

While ring-style sequence parallelism processes only subsequences per transmission and cannot perform global layout transformations, Ulyssesstyle sequence parallelism (Jacobs et al., 2024) is ideally suited for this scenario. Each device initially stores a subsequence of the input with shape $[B, S/N, H, D]$, where $B$ is the batch size, $S$ is the full sequence length, $H$ is the number of attention heads, and $D$ is the head dimension. An all-to-all exchange is first performed so that each device reconstructs full sequences with shape $[B, S, H/N, D]$. With the complete sequence available locally, we can then apply a sparse attention pattern independently to the subset of heads assigned to each device. After the attention computation, a second all-to-all operation restores the tensor layout to $[B, S/N, H, D]$. While the total communication volume remains the same as in conventional Ulysses sequence parallelism, the use of sparse attention greatly reduces the computational cost.

## 4.3 DYNAMIC ATTENTION SCHEDULING

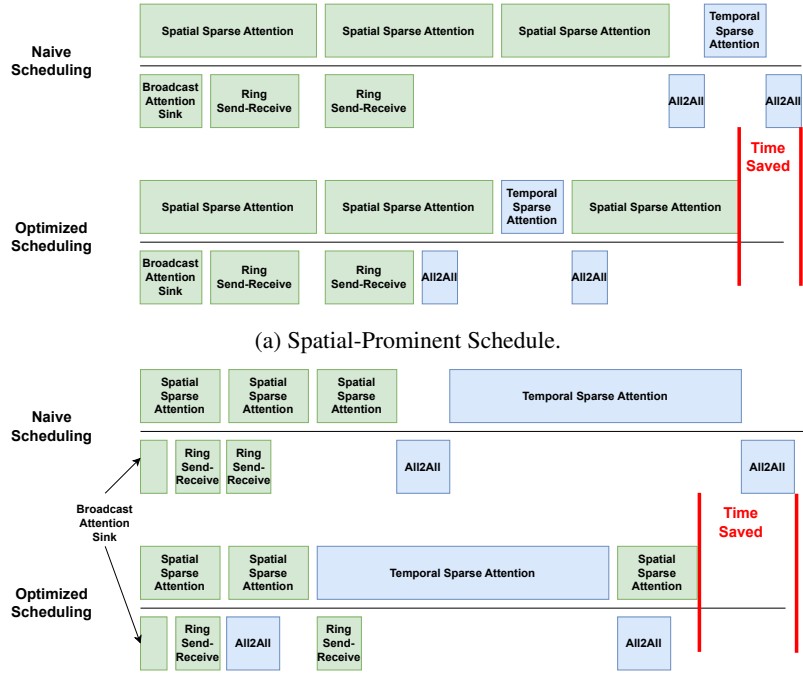

(a) Spatial-Prominent Schedule.

(b) Temporal-Prominent Schedule.

Figure 5: Dynamic attention scheduling. The green boxes represent spatial attention and blue boxes represent temporal attention.

Diffusion models exhibit dynamic behavior in their attention computation across different layers and denoising steps when processing various prompts. Consequently, the ratio between spatial sparse heads and temporal sparse heads fluctuates dynamically throughout the inference process. To enhance performance, we propose dynamic attention scheduling, which efficiently coordinates computation and communication operations. Figure 5 shows the mechanism of our proposed solution.

**Spatial-dominant Schedule.** When spatially sparse heads comprise the majority of attention heads, we interleave spatial attention computation with temporal attention computation. The key optimization is to hide the communication overhead of all-to-all operations through this interleaving strategy.

**Temporal-dominant Schedule.** When temporally sparse heads are dominant, we adopt a different approach. First, we compute the local attention for spatial heads while overlapping this computation with all-to-all communication. During the subsequent Ulysses attention computation, we perform partial-ring communication to gather spatial tokens, which are then concatenated into a larger tensor. Finally, we execute spatial attention computation while simultaneously overlapping it with temporal all-to-all communication.

## 5 EVALUATION

### 5.1 EXPERIMENT SETUP

We evaluated DSA on three state-of-the-art video generation models: Wan2.1-1.3B, Wan2.1-14B, and Hunyuan-Video. We employed VBench (Huang et al., 2024) as our primary benchmark for assessing video quality. Since the original prompts in VBench are concise and limited in complexity, we refined them using GPT-4-mini. From VBench's comprehensive evaluation framework, we selected four critical dimensions: overall consistency, subject consistency, spatial relationship, and temporal style, which together provide a holistic assessment of video generation quality. Additionally, we conducted frame-to-frame comparisons using traditional image quality metrics, including Peak

| Model | Method | Generated Video Quality | | | | | | |
|-------|--------|--------|--------|---------|-------------------------|------------------------|----------------------------|------------------|
| | | PSNR ↑ | SSIM ↑ | LPIPS ↓ | Overall Consistency ↑ | Subject Consistency ↑ | Spatial Relationship ↑ | Temporal Style ↑ |
| Wan2.1-1.3B | Dense | - | - | - | 0.168 | 0.922 | 0.819 | 0.156 |
| | Sparge | 31.39 | 0.704 | 0.175 | 0.166 | 0.909 | 0.713 | 0.152 |
| | SVG | 34.74 | 0.832 | 0.073 | 0.168 | 0.921 | 0.825 | 0.154 |
| | DSA (Ours) | 34.67 | 0.833 | 0.073 | 0.166 | 0.922 | 0.824 | 0.152 |
| Wan2.1-14B | Dense | - | - | - | 0.170 | 0.927 | 0.798 | 0.163 |
| | Sparge | 30.79 | 0.641 | 0.189 | 0.161 | 0.892 | 0.701 | 0.155 |
| | SVG | 33.03 | 0.781 | 0.109 | 0.170 | 0.925 | 0.804 | 0.166 |
| | DSA (Ours) | 33.19 | 0.775 | 0.103 | 0.171 | 0.922 | 0.804 | 0.165 |
| Hunyuan-video | Dense | - | - | - | 0.165 | 0.940 | 0.614 | 0.158 |
| | Sparge | 32.19 | 0.762 | 0.141 | 0.160 | 0.930 | 0.584 | 0.143 |
| | SVG | 33.32 | 0.810 | 0.120 | 0.168 | 0.938 | 0.637 | 0.152 |
| | DSA (Ours) | 33.40 | 0.804 | 0.121 | 0.167 | 0.940 | 0.633 | 0.149 |

Table 1: Video quality evaluation on VBench

Signal-to-Noise Ratio (PSNR), Structural Similarity Index Measure (SSIM), and Learned Perceptual Image Patch Similarity (LPIPS) (Zhang et al., 2018). These evaluation metrics comprehensively cover both image quality and spatial-temporal coherence at the video level.

We compared DSA with both sparse attention and distributed inference approaches. For video quality evaluation, we selected SparseAttention (Zhang et al., 2025a), and Sparse-Video-Gen (SVG) (Xi et al., 2025). Ring/Ulysses Sequence parallelism is not used for quality evaluation as it achieves the same performance as the full attention baseline. For system performance, we compared the generation latency for both sparse attention and distributed methods including SVG (Xi et al., 2025) and SP-Unified (Fang & Zhao, 2024; Gu et al., 2024). As USP enables different combinations of ring and Ulysses attention, we only kept the best results in Table 2.

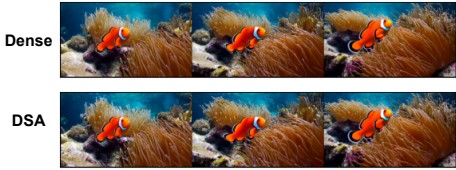
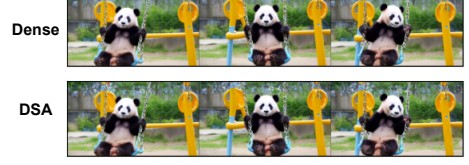

(a) Prompt: A vibrant orange-and-white clownfish darts through a sunlit coral reef, weaving gracefully among swaying anemones and colorful corals.

(b) Prompt: A fluffy panda joyfully swings back and forth on a brightly colored playground swing set.

Figure 6: Visualization of the generated outputs from Wan2.1-14B

## 5.2 VIDEO QUALITY EVALUATION

We evaluated the quality of the videos generated by different methods, and the results are summarized in Table 1. We set the sparsity level to 75% for both SVG and DSA, while using a similarity threshold of 0.6 and a CDF threshold of 0.98 for Sparse Attention. We do not report results for the USP method, as it is lossless and produces identical results to dense attention.

According to the quantitative evaluation metrics, DSA consistently achieves performance comparable to SVG, while outperforming MInference and SparseAttention across both the Wan and Hunyuan models. This indicates that DSA effectively preserves the fidelity and coherence of generated video sequences despite its use of a sparse attention mechanism.

Figure 6 presents two randomly selected prompts along with the videos generated by each method. For both methods, we show frames sampled from the beginning, middle, and end of each video. Visual inspection indicates that the frames produced by DSA closely resemble those generated using dense attention, preserving high visual fidelity and temporal coherence. Additional frames for more diverse prompts are provided in the Appendix A.1, and full video examples are included in the supplementary materials.

| Model | Method | System Performance | | |
| --- | --- | --- | --- | --- |
| | | # of GPUs | Generation time (s) | Speedup |
| Wan2.1-1.3B | Dense | 1 | 402.34 | 1 |
| | SVG | 1 | 310.14 | 1.29 |
| | USP | 8 | 59.45 | 6.76 |
| | DSA (Ours) | 8 | 54.11 | 7.43 |
| Wan2.1-14B | Dense | 1 | 1889.25 | 1 |
| | SVG | 1 | 1221.34 | 1.55 |
| | USP | 8 | 251.26 | 7.52 |
| | DSA (Ours) | 8 | 175 | 10.79 |
| Hunyuan-video-13B | Dense | 1 | 1790.34 | 1 |
| | SVG | 1 | 1340.40 | 1.34 |
| | USP | 8 | 284.71 | 6.29 |
| | DSA (Ours) | 8 | 189.38 | 9.45 |

Table 2: Latency and speed of different models when generating a 720p 5-second video.

| Model | Strategy | Generation time (s) |
| --- | --- | --- |
| Wan2.1-14B | Naive Schedule | 188.92 |
| | Dynamic Schedule | 180.47 |
| | Spatial Only | 175 |

Table 3: Generation latency when adopting different strategies for attention.

## 5.3 System Performance Evaluation

We also investigated the system performance of DSA. Unlike large language models, which typically emphasize metrics such as time to first token (TTFT) and time per output token (TPOT), video generation models place a higher priority on overall generation latency, as the total runtime spans the scale of minutes rather than seconds. As shown in Table 2, DSA significantly outperforms the single-GPU method and can achieve up to 10.79x speedup. This translates to super-linear scaling on Wan-14B and Hunyuan-13B as the speedup is greater than the proportional increase in the number of GPUs, demonstrating promising cost-effectiveness in large-scale deployment. Compared to the distributed unified sequence parallelism, DSA can still achieve 43% improvement on Wan-14B. However, it is noted that DSA still scales sub-linearly for Wan-1.3B, suggesting that the computation sharding hurts the hardware utilization and reduces the computation efficiency.

## 5.4 Ablation Studies

**Scheduling Strategies.** We evaluated the impact of different scheduling strategies on DSA. Under naïve scheduling, spatial and temporal attention are executed sequentially without overlap. In contrast, Dynamic Attention Scheduling reorders execution based on the spatial–temporal ratio and incorporates computation–communication overlap. As shown in Table 3, this dynamic strategy reduces latency by 4.7%. We further examined a spatial-only strategy, where all attention heads adopt the spatial pattern. This configuration decreases generation latency to 175 seconds—an 8% improvement over naïve scheduling—while incurring negligible impact on video quality (results are put in the appendix).

**Sparsity Selection** In DSA, since computation for spatial and temporal patterns is decoupled, we can assign different sparsity levels to each, unlike SVG, which enforces a uniform sparsity level across both. To evaluate this flexibility, we sampled 20 prompts from each VBench evaluation dimension and used Wan2.1-14B to generate videos under varying sparsity settings. Specifically, we experimented with sparsity levels of 80%, 90%, and 95% for both spatial and temporal dimensions, and assessed their impact on video quality. As shown in Table 4, setting spatial sparsity too high degrades performance: when spatial sparsity is increased from 80% to 95% with temporal sparsity fixed at 95%, the overall consistency score drops from 0.179 to 0.174. However, very high temporal sparsity tends to yield comparable performance. For example, a temporal sparsity of 95% produces results similar to those at lower spatial sparsity levels of 90% or 80%. This reveals that temporal attention patterns are generally more sparse than the spatial patterns. This is because the number of frames is generally smaller than the size of tokens in a single frame. Consequently, for a given query token, the number of key tokens at the same spatial location but across different temporal locations is much smaller than the number of key tokens located within the same or adjacent frames.

| Model | Spatial Sparsity | Temporal Sparsity | Overall Consistency ↑ | Subject Consistency ↑ | Spatial Relationship ↑ | Temporal Style ↑ |
|---|---|---|---|---|---|---|
| Wan2.1-14B | 95% | 95% | 0.174 | 0.916 | 0.941 | 0.135 |
| | | 90% | 0.176 | 0.918 | 0.957 | 0.135 |
| | | 80% | 0.177 | 0.918 | 0.957 | 0.134 |
| | 90% | 95% | 0.178 | 0.915 | 0.952 | 0.138 |
| | | 90% | 0.178 | 0.919 | 0.948 | 0.135 |
| | | 80% | 0.178 | 0.920 | 0.950 | 0.134 |
| | 80% | 95% | 0.179 | 0.917 | 0.944 | 0.138 |
| | | 90% | 0.179 | 0.919 | 0.948 | 0.136 |
| | | 80% | 0.177 | 0.921 | 0.956 | 0.135 |

Table 4: Sensitivity to sparsity levels for spatial and temporal attention respectively.

## 6 RELATED WORK

Diffusion models have been accelerated through several largely orthogonal approaches. One line of work focuses on sparse attention. Although methods such as BigBird (Zaheer et al., 2020), StreamingLLM (Xiao et al., 2024b), DuoAttention (Xiao et al., 2024a), and Native Sparse Attention (Yuan et al., 2025) demonstrate strong performance in large language models, they rely on language-specific attention patterns and do not transfer effectively to diffusion models. More recently, SpargeAttention (Zhang et al., 2025a) dynamically detects sparsity and implements an efficient kernel for acceleration. SVG (Xi et al., 2025) and STA (Zhang et al., 2025c) extend sparse attention to diffusion models by applying static sparsity patterns, among which SVG achieves the best generation quality.

Another line of work focuses on system-level optimization, including DistriFusion (Li et al., 2024) and PipeFusion (Fang et al., 2025). DistriFusion leverages stale latents and patch parallelism to partition images across devices for parallel computation, while PipeFusion extends this approach with pipeline parallelism to further reduce latency and improve hardware utilization. However, these methods primarily target image generation. USP (Fang & Zhao, 2024) instead proposes a lossless distributed inference framework that combines Ring Attention (Li et al., 2023; Liu et al., 2024) and Ulysses (Jacobs et al., 2023) to improve scalability.

A third line of work explores caching mechanisms that reuse intermediate activations based on the similarity of latent representations across denoising steps. PAB (Zhao et al., 2025) and DiTFastAttn (Yuan et al., 2024) use static reuse, while AdaCache (Kahatapitiya et al., 2024) adapts caching based on feature variance and TaylorSeer (Liu et al., 2025) predicts feature evolution via Taylor expansion. These methods are complementary to sparse attention and distributed inference.

In contrast to prior work, our method jointly considers both the sparsity characteristics of attention patterns and distributed inference strategies. By aligning sparsity-aware computation with distributed execution, our approach improves both computational efficiency and communication efficiency. Furthermore, our method is orthogonal to caching-based techniques and can be seamlessly combined with them for additional acceleration.

## 7 CONCLUSION

In conclusion, we introduce DSA, a novel attention mechanism that integrates sparse attention with distributed inference for diffusion-based video generation. By selecting suitable parallel strategy for distinct sparse patterns, DSA substantially reduces computation and communication overhead. Experiments demonstrate that DSA achieves significant efficiency gains: up to 10.79× faster inference compared to the single-GPU dense attention inference while preserving the video quality.

This work explores parallelization strategies for spatial and temporal attention patterns, without yet addressing sparse patterns that may emerge in future models. Although dynamic attention scheduling overlaps computation and communication, its multiple kernel launches can degrade performance. As future work, we plan to incorporate adaptive sparse patterns and fuse compute–communication into efficient CUDA kernels using libraries such as TileLink (Zheng et al., 2025b) and Triton-Distributed (Zheng et al., 2025a).

ACKNOWLEDGMENTS

We thank the anonymous reviewers for their valuable feedback and constructive suggestions. We are grateful to our collaborators and colleagues for insightful discussions and support throughout this project. Shenggui Li is generously supported by the A*STAR ACIS scholarship.

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

# A  APPENDIX

## A.1  VISUAL COMPARISON BETWEEN DSA AND BASELINES

The figures below present videos generated using full attention, SVG (Xi et al., 2025), and our proposed DSA method, respectively. We employ Wan2.1-14B Wan et al. (2025) to generate the videos and extract one frame every 10 frames to illustrate both spatial and temporal consistency.

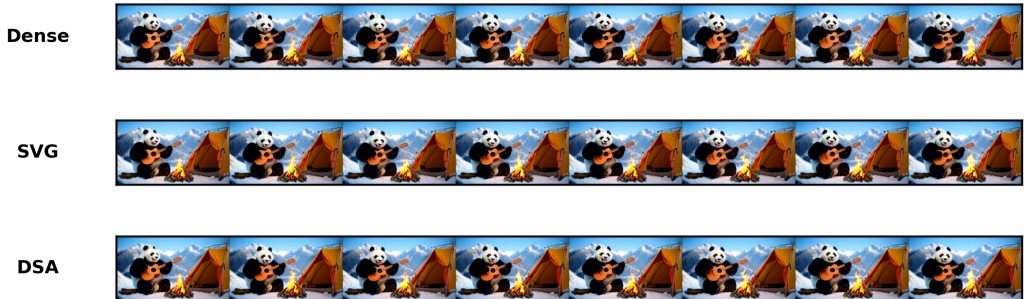

Figure 7: Prompt: A cheerful, fluffy panda strums a guitar beside a crackling campfire, with snow-capped mountains rising in the background.

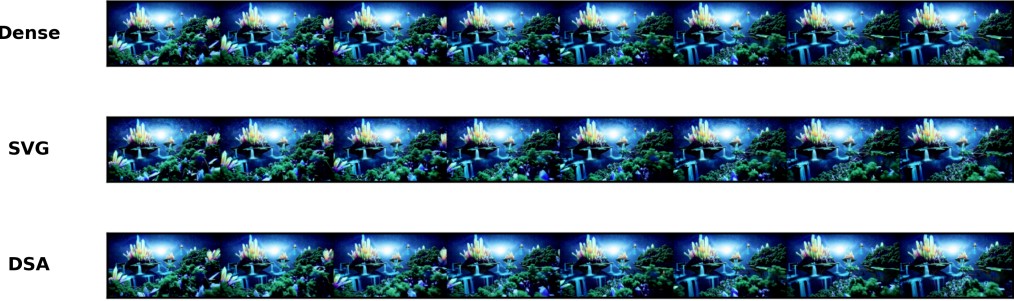

Figure 8: Prompt: A camera soars through surreal fantasy landscapes—floating islands, crystalline spires, bioluminescent forests, cascading waterfalls, and a shifting, star-lit sky.

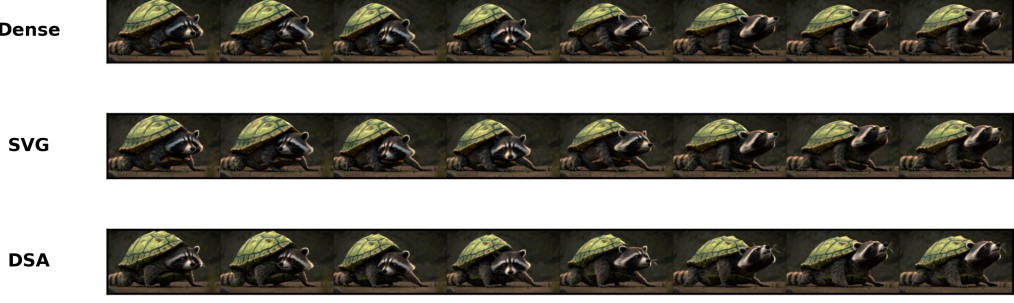

Figure 9: Prompt: Digital-art video of a whimsical hybrid creature: a raccoon with a textured turtle shell and subtle reptilian markings, rendered with detailed fur and shell textures, soft cinematic lighting, and gentle, playful animation.

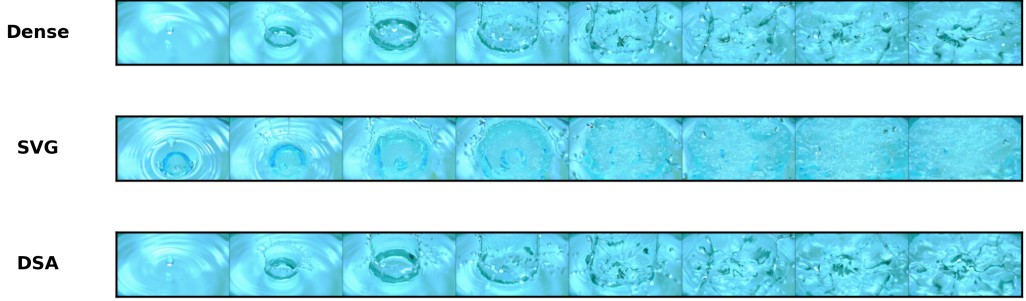

Figure 10: Prompt: Extreme slow-motion close-up of a vibrant turquoise water splash with fine droplet detail on a transparent background (alpha channel included).

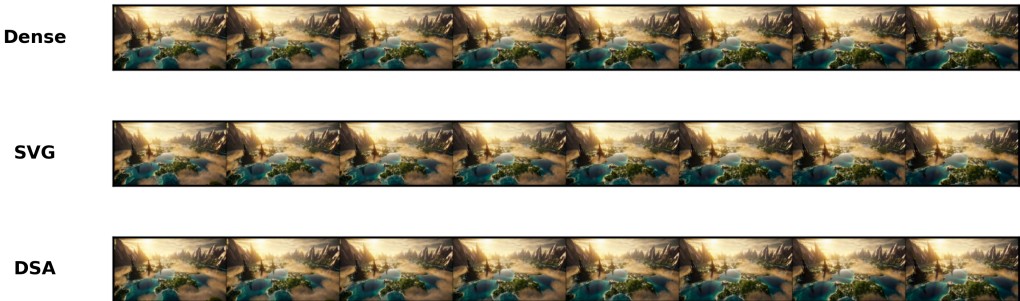

Figure 11: Prompt: Smooth, cinematic aerial panoramic drone shot sweeping over a vivid fantasy realm of floating islands, crystalline lakes, mist-shrouded forests, and towering ancient spires bathed in warm golden-hour light.

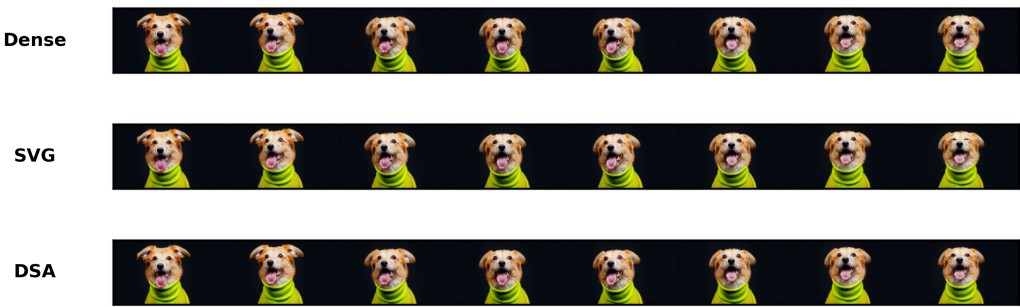

Figure 12: Prompt: Studio portrait of a happy dog facing the camera, wearing a bright yellow turtleneck, centered in frame with soft studio lighting against a dark background.

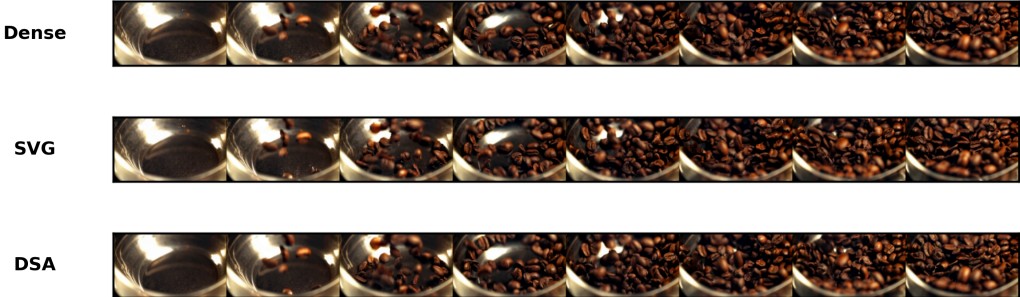

Figure 13: Prompt: Tightly cropped macro slow-motion close-up of roasted coffee beans cascading into an empty bowl, highlighting surface texture, sheen, and the graceful motion of individual beans.

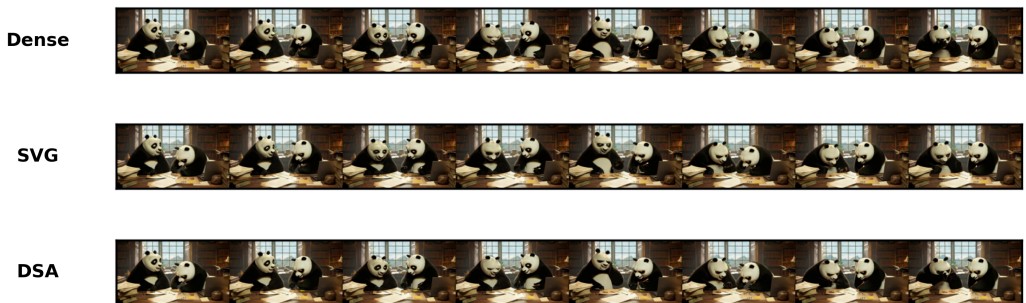

Figure 14: Prompt: Two pandas in a cozy study animatedly discuss an academic paper, pointing at charts, flipping pages, and jotting notes on a cluttered desk.

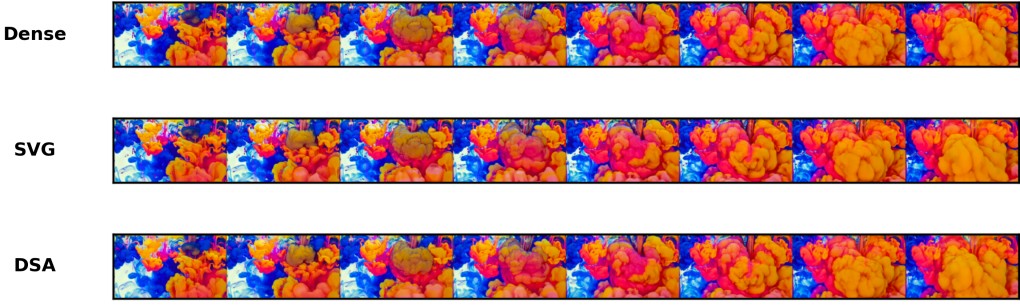

Figure 15: Prompt: Vibrant ink droplets swirl and diffuse through water, forming dreamy, abstract cloud-like color formations.

