# OpenReview forum: "DSA: Efficient Inference For Video Generation Models via Distributed Sparse Attention"
_ICLR.cc/2026/Conference — ICLR 2026 Poster_

### Official Review · Reviewer_RRN5 · 2025-10-30

**Soundness:** 3
**Presentation:** 4
**Contribution:** 3
**Rating:** 4
**Confidence:** 3

**Summary:**

This paper proposes DSA, a system that combines sparse attention mechanisms with distributed inference to accelerate video generation using Diffusion Transformer models. The approach introduces two key components: (1) Mixed Parallelism (MP) that applies different parallelism strategies (spatial sequence parallel vs. temporal sequence parallel) based on attention patterns, and (2) Dynamic Attention Scheduling (DAS) that optimizes computation-communication overlap. Experiments on Wan and Hunyuan-Video models show up to 10.79× speedup over single-GPU inference while maintaining video quality.

**Strengths:**

The paper makes a solid contribution by integrating sparse attention with distributed inference for video generation. The key insight of matching different parallelism strategies to spatial versus temporal attention patterns is well-motivated and novel. The experimental evaluation covers multiple models with comprehensive quality and system performance metrics, demonstrating impressive super-linear scaling for larger models. The training-free nature makes it immediately applicable to existing models.

**Weaknesses:**

1.The paper only reports PSNR, SSIM, and LPIPS metrics without perceptual quality metrics like VBench scores that evaluate specific video generation dimensions such as subject consistency, temporal style, spatial relationships, and overall consistency. These metrics are crucial for assessing whether sparse patterns preserve the semantic and temporal coherence of generated videos.

2.The evaluation does not compare against other sparse attention methods for video generation such as DiTFastAttn, MInference applied to video models, or cache-based methods like PAB.

3.Critical design decisions lack ablation studies. The paper does not validate whether the mixed parallelism strategy outperforms a uniform strategy. The impact of dynamic scheduling versus naive scheduling is mentioned briefly but not thoroughly evaluated across different models and workloads. There is no sensitivity analysis on the sparsity hyperparameters (cs and ct) to show robustness across different efficiency-accuracy trade-offs.

4.The paper provides only two visual examples in Figure 6. More extensive qualitative comparisons across diverse scenarios (minor vs. significant scene changes, rare vs. frequent object interactions) would strengthen the quality claims, especially given the method's high PSNR but potential for subtle temporal artifacts.

5.While the paper claims hardware efficiency through layout transformation, it does not provide detailed kernel-level benchmarks comparing the proposed approach against naive sparse attention implementations across different sparsity levels.

**Questions:**

See weakness

---

> ### Author Response · Authors · 2025-11-26
> **Response to Reviewer RRN5**
>
> Thank you for your time and thorough review, we have answered your questions below and hope that they will address your concerns.
> 1. Besides the traditional metrics like PSNR, SSIM and LPIPS, We presented in both Table 1 and 4 the VBench scores on the subject consistency, temporal style, spatial relationships, and overall consistency.
> 2. DSA demonstrates comparable performance to SVG and SVG outperforms methods such as DiTFastAttn, MInference and PAB as given in [1].
> 3. We presented the performance gains obtained from mixed scheduling compared to naive scheduling in Table 3. The performance gain is obtained from better overlapping of computation and communication based on the spatial-temporal ratio as shown in Figure 5.
> 4. I have uploaded more videos in the supplimentary materials for reference. Meanwhile, the appendix section contains more samples for different prompts.
> 5. During processing the temporal attention, we applied layout transformation from token-major to frame-major as done in the SVG paper [1].
>
> [1] Sparse VideoGen: Accelerating Video Diffusion Transformers with Spatial-Temporal Sparsity

---

### Official Review · Reviewer_HyRK · 2025-10-31

**Soundness:** 2
**Presentation:** 2
**Contribution:** 2
**Rating:** 4
**Confidence:** 5

**Summary:**

This paper proposes Dynamic Selective Attention (DSA) to accelerate video generation.

DSA adopts the dynamic sparse patterns based on the fixed spatial and temporal sparse patterns.

This work mainly focus on the contribution on the system level scheduling design.

The experimental results show that this work achieves good results.

**Strengths:**

1. This work shows how to schedule the attention computation for sparse attention.

**Weaknesses:**

1. This work does not provide the detailed dynamic sparse pattern online generation methods. In Section 4.1, this work only shows that they adopted the spatial and temporal kernels from the work [1], which shows the limited novelty. The dynamic pattern looks like the token-importance based pruning, which is not novel.

2. This work claims dynamic sparse pattern, while it compares to the fixed sparse pattern works like [1]. This work does not compare to dynamic sparse pattern works like [2]


---
[1] Sparse VideoGen: Accelerating Video Diffusion Transformers with Spatial-Temporal Sparsity

[2] DraftAttention: Fast Video Diffusion via Low-Resolution Attention Guidance

**Questions:**

1. Please provide the details of the dynamic sparse pattern generation method.

2. Please provide the overhead (like latency) brought by the dynamic sparse pattern generation.

3. Please provide the comparison to dynamic sparse pattern works in video generation like [1].

---
[1] DraftAttention: Fast Video Diffusion via Low-Resolution Attention Guidance

---

> ### Author Response · Authors · 2025-11-26
>
> Thank you for your time and thorough review, we have answered your questions below and hope that they will address your concerns.
> 1. In DSA, we did not use dynamic sparse attention, instead, we follow SVG's definition of temporal and spatial attention patterns. During runtime, we dynamically schedule the attention comptuation based on the relative ratio of spatial-to-temporal heads.
> 2. As mentioned in 1, there is no overhead brought by dynamic sparse attention. The only overhead is that we have to perform majority voting to decide the attention pattern for each head.
> 3. As we are not using dynamic sparse attention, we did not compare it with DraftAttention. However, in our paper, we still included the comparison with the current well-known dynamic sparse attention methods like SpargeAttention.

---

### Official Review · Reviewer_3sqw · 2025-11-01

**Soundness:** 2
**Presentation:** 3
**Contribution:** 3
**Rating:** 6
**Confidence:** 5

**Summary:**

The paper presents an interesting approach to improving the efficiency of video generation models using distributed sparse attention. However, the lack of supplementary materials, incomplete evaluation, and inconsistent reporting of results hinder the ability to fully assess the effectiveness of the proposed method. Addressing these issues will significantly enhance the paper's quality and credibility.

**Strengths:**

The paper presents an interesting approach to improving the efficiency of video generation models using distributed sparse attention.

**Weaknesses:**

1. The authors have not provided the MP4 files for the proposed method and comparison methods in the supplementary materials. This makes it very difficult to assess the actual visual quality and temporal consistency of the generated videos.

2. Even without the MP4 files, the authors could have provided code or other means to reproduce the results and evaluate the visual quality.

3. The authors have not tested the full range of metrics from VBench or VBench2.0. Comprehensive evaluation is crucial to understand the strengths and weaknesses of the proposed method.

4. Table 2 includes timing results for USP, but Table 1 lacks corresponding quality metrics for USP. This inconsistency makes it difficult to compare the proposed method with USP comprehensively.

5. The related work section could benefit from a more comprehensive review of pre-trained models for video generation acceleration and sparse/linear attention methods.

If the authors address the above concerns effectively, particularly by providing supplementary materials and a more comprehensive evaluation, I would be willing to reconsider my assessment and potentially give a more positive score.

**Questions:**

no

---

> ### Author Response · Authors · 2025-11-26
>
> Thank you for your time and thorough review, we have answered your questions below and hope that they will address your concerns.
> 1. I have added some MP4 files in the supplementary materials and hope that these can better help you assess the video quality. Due to the file size limit, I can only upload 6 videos per dimension. Meanwhile, in the appendix section of the paper, more samples are provided in the form of screenshots.
> 2. Same as above
> 3. 1. The authors have not tested the full range of metrics from VBench or VBench2.0. Comprehensive evaluation is crucial to understand the strengths and weaknesses of the proposed method.
> 4. USP is a lossless method as it does not change the way how model computes attention, it just parallelizes the computation. Therefore, Table 1 does not present evaluation metrics for USP as it is actually the same as that of the dense model.
> 5. Sure, we will add more relevant content on these two aspects by covering models including CogVideo, OpenSora, OpenSora-Plan, HunYuan, Wan, Mochi and attention methods such as STA, SVG etc. We will also discuss some attention methods suitable for LLMs but not video models.

---

> ### Comment · Reviewer_3sqw · 2025-11-28
>
> I appreciate the authors’ thorough responses. They have addressed most of my concerns. I am more than happy to maintain my positive score.

---

### Official Review · Reviewer_FLsG · 2025-11-01

**Soundness:** 2
**Presentation:** 2
**Contribution:** 2
**Rating:** 4
**Confidence:** 3

**Summary:**

This paper proposed distributed sparse attention (DSA) for DiT-based video generation model and achieved $1.43 \times$ speed up than the unified sequence parallelism (USP).

**Strengths:**

1. By combining sparse attention with distributed strategies, the video generation model achieves improved multi-GPU efficiency while preserving the original generation quality as much as possible.

2. Evaluations were conducted on several mainstream models, including Wan2.1-1.3B, Wan2.1-14B, and Hunyuan-Video, covering video quality metrics (PSNR, SSIM, LPIPS, VBench) as well as system performance metrics (latency and speedup).

**Weaknesses:**

1. The overall approach resembles a combination of the SVG method and the USP method, with optimized attention scheduling.

typo: Line 152, "xx tokens"

**Questions:**

1. In Table 2, the results of USP + SVG can be provided.

---

> ### Author Response · Authors · 2025-11-26
>
> Thank you for your time and thorough review, we have answered your questions below and hope that they will address your concerns.
> 1. As USP runs ulysses first followed by ring attention, SVG is not directly compatible with it because it does not have a whole sequence on a single GPU during ring computation.

---

### Comment · Area_Chair_soby · 2025-11-27

Dear Reviewers,

This is a gentle reminder to please take a moment to review the author rebuttals and check whether your main concerns have been adequately addressed.

If possible, please update your reviews or add a brief clarification on whether the responses resolved your questions or if any issues remain. Your follow-up feedback is important for ensuring a fair and well-informed decision process.

Thank you again for your time and for helping maintain the quality of the ICLR review process.

Best,
AC

---

### Meta-Review · Area_Chair_oboo · 2026-01-08

**Summary:**

The reviewers express comments from the following perspectives.

1. Technical novelties. Approach similar with the combination of SVG ang USP. [Reviewer FLsG]
2. Insufficient experiments. No generated video files [Reviewer 3sqw, RRN5], Vbench and Vbench2.0 evaluation [Reviewer 3sqw, RRN5], more competitor models [Reviewer RRN5], lacking ablation studies [Reviewer RRN5].
 3. Writing Improvement [Reviewer 3sqw].

Most of the review comments focus on the evaluations, which have been addressed. And the writing issues, specifically the related work can be easily addressed. The technical contributions seems to be not so significant but is not one work on simple combination of SVG and USP.

Considering this, I am suggesting the acceptance of the paper.

**Reviewer Concerns:**

The insufficient evaluation and writing improvement are well addressed.

**Reviewer Scores:**

Reviewers 3sqw, RRN5 who mainly concern the insufficient experiments may provide positive score.

---

### Decision · Program_Chairs · 2026-01-26

Accept (Poster)